# Low-Threshold and High-Extinction-Ratio Optical Bistability within a Graphene-Based Perfect Absorber

**DOI:** 10.3390/nano13030389

**Published:** 2023-01-18

**Authors:** Zhengzhuo Zhang, Qiaoge Sun, Yansong Fan, Zhihong Zhu, Jianfa Zhang, Xiaodong Yuan, Chucai Guo

**Affiliations:** College of Advanced Interdisciplinary Studies & Hunan Provincial Key Laboratory of Novel Nano-Optoelectronic Information Materials and Devices, National University of Defense Technology, Changsha 410073, China

**Keywords:** graphene, perfect absorber, optical bistability, nonlinear optics, nanostructure

## Abstract

A kind of graphene-based perfect absorber which can generate low-threshold and high-extinction-ratio optical bistability in the near-IR band is proposed and simulated with numerical methods. The interaction between input light and monolayer graphene in the absorber can be greatly enhanced due to the perfect absorption. The large nonlinear coefficient of graphene and the strong light-graphene interaction contribute to the nonlinear response of the structure, leading to relatively low switching thresholds of less than 2.5 MW/cm^2^ for an absorber with a *Q* factor lower than 1000. Meanwhile, the extinction ratio of bistable states in the absorber reaches an ultrahigh value of 47.3 dB at 1545.3 nm. Moreover, the influence of changing the structural parameters on the bistable behaviors is discussed in detail, showing that the structure can tolerate structural parametric deviation to some extent. The proposed bistable structure with ultra-compact size, low thresholds, high extinction ratio, and ultrafast response time could be of great applications for fabricating high-performance all-optical-communication devices.

## 1. Introduction

Optical bistability (OB) is a typical phenomenon of nonlinear optics. In some resonant structures with third-order nonlinear optical materials, one given input state has two corresponding output states. Based on this unique property, OB can be applied to the fabrication of all-optical switches, memory cells, and transistors [1,2,3,4,5,6], which are the fundamental components of all-optical communication systems. Up to now, researchers have developed many different methods to realize OB, including ring cavities [7,8], photonic crystals [9,10], subwavelength waveguides [11,12] etc. However, the third-order susceptibilities of bulk dielectrics are usually very small, and as a result, either ultrahigh intensity of incident light or an ultrahigh *Q* factor is required to generate bistable states, which limits the applications of OB. Therefore, materials with large nonlinear coefficients are needed to overcome this drawback.

Graphene, as a novel 2D material, has been studied intensively over the years due to its outstanding physical properties [13,14], such as full-spectrum optical response, ultrafast response speed, ultrahigh carrier mobility, controllable thermal emission, etc. At the same time, the nonlinear response of graphene is also intriguing since it is much stronger than that of conventional nonlinear materials. It is reported that graphene possesses a giant nonlinear refractive index of more than 10^−12^ or even 10^−11^ m^2^/W, which is about 10^7^ to 10^9^ times larger than that of bulk dielectrics [15,16,17]. Such an excellent characteristic makes it an ideal material for nonlinear optical applications. However, the interaction between light and monolayer graphene is very weak due to the ultrathin thickness of graphene, which seriously limits the applications of graphene in the optical field. In order to greatly enhance the light-graphene interaction, many kinds of graphene-based resonators [18,19,20,21,22,23,24,25,26] and perfect absorbers [27,28,29,30,31,32,33,34] have been proposed, and some applications of those structures have been demonstrated [35,36,37,38,39]. Until now, OB in graphene-based resonators has been theoretically and numerically studied [40,41,42,43,44,45,46,47,48,49,50,51,52,53,54]. Most of the previous studies focus on the mid-IR to terahertz band since low-threshold OB could be achieved in those structures due to the strong field enhancement caused by the surface plasmons of graphene. In the visible to near-IR band, however, graphene plasmonic resonances can hardly be excited due to the limit of the doping level [55,56], and graphene-based OB with low thresholds in this band is still challenging.

In this article, we demonstrate graphene-based OB with a low threshold and high extinction ratio by using a graphene-based perfect absorber. The absorber supports guided-mode resonance (GMR) in the near-IR band, which strongly enhances the electric field intensity in the graphene. With the significant light-graphene interaction as well as the large nonlinear coefficient of graphene, the switching threshold of the bistable states is cut down to a relatively low level. Moreover, by optimizing structural parameters to satisfy the critical coupling condition, the extinction ratio of bistable states reaches an ultrahigh level. As is demonstrated by the simulation, the switching threshold can be lower than 2.5 MW/cm^2^, and the extinction ratio is over 40 dB. The proposed structure is easy to fabricate and has a toleration to structural parametric deviation, making it practical for the real applications of high-performance graphene-based bistable devices.

## 2. Model and Methods

Figure 1 shows the schematic of our proposed graphene-based perfect absorber. Monolayer graphene is sandwiched between a 1D polymethy1-methacrylate (PMMA) grating and a silica layer deposited onto a gold substrate. The grating layer in the absorber is essential for generating GMRs, which can greatly enhance the light-graphene interaction. For a designed absorber, the grating period *p* = 1138 nm, the grating thickness *h =* 67 nm, and the grating width *w* = *p*/2. The thickness of the silica layer *d* = 1200 nm, and the thickness of the gold substrate is 200 nm. The thickness of the graphene $t_{g}$ = 0.34 nm, the refractive indices of PMMA and silica are set to 1.48 and 1.45, respectively, with their dispersion effect neglected. The gold layer is modeled as a Drude material which can be expressed by $\varepsilon (\omega )=\varepsilon_{\infty }\;-\;{\omega_{p}}^{2}/(\omega_{2}+i\gamma \omega )$, where $\varepsilon_{\infty }$ = 1.0, $\omega_{p}$ = 1.37 × 10^16^ s^−1^ and *γ* = 8.17 × 10^13^ s^−1^ [29]. The intraband and interband conductivity of graphene is described by the Kubo formula [57]:(1)$$\sigma_{intra}(\omega )=\frac{i2e^{2}k_{B}T}{\pi \hbar^{2}(\omega +i\tau^{-1})}\ln\left[2\cosh\left(\frac{E_{f}}{2k_{B}T}\right)\right]$$
(2)$$\sigma_{inter}(\omega )=\frac{e^{2}}{4\hbar^{2}}\left[\frac{1}{2}+\frac{1}{\pi }\arctan\left(\frac{\hbar \omega -2E_{f}}{2k_{B}T}\right)-\frac{i}{2\pi }\ln\frac{{\left(\hbar \omega +2E_{f}\right)}^{2}}{{\left(\hbar \omega -2E_{f}\right)}^{2}+{\left(2k_{B}T\right)}^{2}}\right]$$
where *E_f_* is the Fermi level, *τ* is the relaxation time and *T* is the temperature. In the simulation of this work, *E_f_* is set to 0.1 eV, considering that graphene might be slightly doped even if gate voltage is absent [15]. *τ* and *T* are set to 200 fs and 300 K, respectively, which are typical values for graphene relaxation time and environmental temperature. The total surface conductivity of graphene is ${\sigma (\omega )=\sigma }_{\mathrm{intra}}{+\sigma }_{\mathrm{intra}}$, and the relative permittivity of graphene is described by $\varepsilon_{r}{=1+i\sigma (\omega )/\varepsilon }_{0}{\omega t}_{g}$, where $\varepsilon_{0}$ is the vacuum permittivity, and $t_{g}$ is the thickness of monolayer graphene. The optical Kerr effect of graphene can be expressed as $\varepsilon_{r}{}^{\prime }{(\omega )=\varepsilon }_{r}\left(\omega \right){+\chi }^{\left(3\right)}{|E}_{\mathrm{loc}}{|}^{2}$, where $\varepsilon_{r}{}^{\prime }\left(\omega \right)$ is the relative permittivity of graphene under an electric field and ${|E}_{\mathrm{loc}}|$ is the amplitude of the local electric field. $\chi^{\left(3\right)}$ is the third-order susceptibility of graphene, and it can be calculated by $n_{2}{=3Z}_{0}\chi^{\left(3\right)}{/4n}_{0}{}^{2}$, where $Z_{0}=377\;\Omega$ is the vacuum impedance, and $n_{2}$ is the nonlinear refractive index of graphene [17]. In this work, we assume that $n_{2}\;\sim \;10^{-12}$ m^2^/W, which is a moderate value for Z-scan measurement in the near-IR band [15,16], and the corresponding nonlinear susceptibility is $\chi^{\left(3\right)}$ ≈ 3 × 10^−14^ m^2^/V^2^. The nonlinear coefficient of graphene is several orders larger than that of bulk silica, so here silica can be treated as a linear dielectric. Besides the optical Kerr effect, we also consider the saturable absorption effect of graphene, which is necessary for intrinsic or lowly doped graphene [15]. The absorption coefficient of graphene can be fitted by ${\alpha (I)=\alpha }_{s}{/(1+I/I}_{s}{)+\alpha }_{\mathrm{ns}}$, where $\alpha_{s}$ and $\alpha_{\mathrm{ns}}$ are the saturable and non-saturable components of the absorption coefficient, and $I_{s}$ is the saturation intensity. Here we assume that $I_{s}$ = 74 MW/cm^2^ and the modulation depth of saturable absorption is $\Delta {=\alpha }_{s}{/(\alpha }_{s}{+\alpha }_{\mathrm{ns}})=74\%$, according to reported measurements [15]. The sum of $\alpha_{s}$ and $\alpha_{\mathrm{ns}}$ equals to α(0), which is determined by the conductivity of graphene (Equations (1) and (2)). By fitting the image part of graphene’s refractive index *k* (which can be derived from the conductivity of graphene) with $k(I)=k(0)\;\cdot \;(\Delta {/(1+I/I}_{s})+1\;-\;\Delta )$, we can equivalently simulate the saturable absorption effect of graphene. The local optical intensity *I* is treated as time-average power flow and is calculated by *I* = |*E*|·|*H*|/2. Apparently, the saturable absorption effect of graphene leads to a decrease in graphene’s absorption coefficient with increasing incident intensity, which should be carefully considered when optimizing structural parameters.

The model of the proposed structure is defined and simulated in the FEM software COMSOL Multiphysics. In the simulation, a 2D model of the structural cross-section in the *xy* plane is built since the structure is uniform along the *z* direction. Periodic boundary condition is added to the *x* direction of the model so that only one structural period needs to be simulated. A port is added above the grating to give a normal plane wave incidence with a certain intensity and scattering boundary condition is added below the gold layer.

## 3. Results and Discussion

Figure 2a demonstrates the reflectance and absorption spectrums from 1535 to 1550 nm under low incident intensity. The demonstrated absorption peak is caused by the zeroth-order GMR with TE polarized normal incidence (parallel to the gratings). The total reflectance and absorption of the structure reach 12.3% and 87.7%, respectively, at the resonant wavelength of 1542.9 nm. The peak absorption of graphene reaches 77.9%, much stronger than that of suspended monolayer graphene, indicating a significant enhancement of light-graphene interaction. It is important to note that the peak absorption of the structure has not reached 100% yet, since herein, the mode leakage rate of the incident port $\gamma_{1}$ is lower than the structural absorption rate $\gamma_{a}$, according to the coupled mode theory [36]. As we will see later, to achieve better bistable behaviors, perfect absorption (requiring that $\gamma_{1}=\gamma_{a}$, which is called the critical coupling condition) is going to be realized under stronger incidence, which makes OB obvious. Figure 2b shows the distribution of the normalized electric field at 1542.9 nm. For TE resonant modes, the electric field does not possess *x* or *y* component, so only the *z* component of normalized electric field (${|E}_{z}{|/|E}_{0}|$) is shown in the graph. Apparently, the electric field is well confined in the silica layer and is enhanced by 15.4 times at the central part of the silica layer compared with the incident electric field. The field intensity inside the graphene is also relatively strong and is enhanced by 8.7 times right under the PMMA ribbon. Figure 2c shows the reflectance spectrums with different *E_f_* under low incident intensity. As *E_f_* increases, the peak absorption of the structure increases and then decreases. This is because the absorption coefficient of graphene, as well as the structural absorption rate $\gamma_{a}$ tends to decrease with *E_f_* increasing. As we can see, when *E_f_* increases to 0.4 eV, the peak absorption reaches almost 100%, indicating that $\gamma_{1}\;\approx \;\gamma_{a}$. As *E_f_* further increases, $\gamma_{a}$ becomes smaller than $\gamma_{1}$, making the peak absorption drop back. Figure 2d shows the reflectance spectrums with different *τ* under low incident intensity. Herein *E_f_* = 0.1 eV. The reflectance spectrum is almost invariant with *τ* changing because in the near-IR band, the conductivity of graphene is mainly determined by the interband transition, which is not influenced by *τ*.

It is known that the rise of incident intensity has a trend to break the spectral symmetry of a resonant structure with Kerr material, with the resonant wavelength shifting away from the initial value and the spectral curve becoming steeper and steeper on one side of the resonant wavelength. Figure 3a shows the nonlinear reflectance spectrums under several incident intensities. When the incident intensity is weak (1 kW/cm^2^), the reflectance spectrum almost coincides with the linear one. When the incident intensity is strong enough (1.5 and 1.75 MW/cm^2^), we see a red shift of the resonant wavelength which is caused by the shift of graphene’s relative permittivity, and meanwhile, the reflectance suddenly jumps up as the incident wavelength is swept over the resonant wavelength from left to right, indicating the appearance of bistable states.

To obtain the hysteric behaviors of the perfect absorber, we should choose a working wavelength that deviates from the initial resonant wavelength, and what is more, the incident intensity $I_{\mathrm{in}}$ should be swept increasingly and then decreasingly with the calculation of the next step taking the result of the last step as the initial condition. This procedure is equivalent to setting the incident intensity as a staircase function within FDTD calculations [40]. Figure 3b shows several hysteresis loops with their working wavelengths chosen to be the same as the resonant wavelengths of the spectral curves in Figure 3a with the same colors and types (except for the black solid curve). The higher branches of the loops are calculated with $I_{\mathrm{in}}$ increasing and the lower branches with $I_{\mathrm{in}}$ decreasing. The hysteresis loops keep expanding as the working wavelength moves away from the initial resonant wavelength, and bistable states become increasingly obvious. The switching-on-to-off (switching-off-to-on) thresholds are 1.7 (1.43) and 2.48 MW/cm^2^ (1.68 MW/cm^2^) for 1545.1 and 1545.6 nm, respectively. These relatively low thresholds are the main contributions of the strong GMR as well as the large nonlinear coefficient of graphene. The simulation results show that when the incident wavelength is between 1545.1 and 1545.6 nm, the extinction ratio of bistable states is greater than 30 dB, and when the wavelength is 1545.3 nm, the extinction ratio reaches the maximum value of 47.3 dB. As mentioned before, the absorption coefficient of graphene has a trend to decrease with increasing light intensity, so the critical coupling condition ($\gamma_{1}=\gamma_{a}$) will be satisfied at certain incident intensities. As a result, incident light will be totally absorbed by the absorber at the off state, and the extinction ratio of the bistable states will ascend to an ultrahigh level. Figure 3c shows the distribution of the normalized electric field of the bistable states at 1545.6 nm with $I_{\mathrm{in}}$ = 1.75 MW/cm^2^. At the on state, reflection is relatively strong since the structure is off-resonant, while at the off state, the structure becomes on-resonant and, since it is critically coupled, the incident light is totally absorbed by the absorber.

In Table 1, we compare the results of Figure 3 with previous works on OB within graphene-based absorption structures. Apparently, the proposed graphene-based absorber has the advantage of both a low threshold and high extinction ratio compared with other works on graphene-based OB in the near-IR band. We can also see that the threshold of graphene-based OB in the THz band can be several orders lower than that in the near-IR band, mainly due to the much stronger resonance effect and field confinement aroused by the surface plasmons of graphene.

We expect the proposed structure to be practical for real experiments, so it is necessary to discuss the influences of structural parametric deviation on its bistable behaviors, considering the possible inaccuracy of fabrication equipment.

Figure 4a shows the variation of nonlinear reflectance spectrums with the silica layer thickness *d* varying from 1170 to 1230 nm with a step of 10 nm. Other structural parameters are the same as those in Figure 2 and Figure 3, and meanwhile, $I_{\mathrm{in}}$ is fixed to 1.75 MW/cm^2^ to make a comparison with the magenta curve in Figure 3a. From the graph, we see that the resonant wavelength blue shifts as *d* decreases and vice versa, which can be explained by the optical coherence theory. The shift of the resonant wavelength with the deviation of *d* is approximately linear (1.2 per 10 nm) within the simulation range. The peak absorption of the structure shows better stability to the deviation of *d* compared with the resonant wavelength. As is shown in the graph, the structure maintains a considerable peak absorption of over 98% within the simulation range, which means that the extinction ratio of bistable states at 1.75 MW/cm^2^ is very stable, even if *d* deviates 30 nm away from the optimized value. Figure 4b shows the hysteresis loops with *d* = 1170, 1200, and 1230 nm at their corresponding working wavelengths, which are 1541.7, 1545.6, and 1549.2 nm, respectively. Bistable characteristics are clear enough for all cases. Apparently, the switching-off-to-on thresholds are maintained at nearly 1.7 MW/cm^2^, and the extinction ratios at 1.75 MW/cm^2^ are still very high, which are 18.8 and 19.7 dB for *d* = 1170 and 1230 nm. The switching-on-to-off threshold decreases with *d* increasing, which has a trend to slightly cut down the structural *Q* factor.

Figure 5a shows the shift of nonlinear reflectance spectrums with the grating thickness *h* varying from 52 to 82 nm with a step of 5 nm. Other structural parameters are still the same as those in Figure 2 and Figure 3, and $I_{\mathrm{in}}$ is also fixed to 1.75 MW/cm^2^. The change of *h* has only a slight influence on the resonant wavelength, which shifts no more than 1.5 nm within the simulation range. Its influence on the peak absorption is much stronger compared with that of the silica layer thickness. The peak absorptions drop to 93.4% and 95.7% for *h* = 52 and 82 nm, respectively. Moreover, the jump of reflectance at the right side of the resonant wavelength is weakened with *h* increasing, indicating that the structural bistable behaviors will be affected. Figure 5b shows the hysteresis loops with *h* = 52 nm, 67 and 82 nm at their corresponding working wavelengths, which are 1544.3, 1545.6, and 1546.5 nm, respectively. The extinction ratios at 1.75 MW/cm^2^ are 13.5 and 13.7 dB for *h* = 52 and 82 nm, respectively. Apparently, the hysteresis loop sharply shrinks with *h* increasing since *h* directly determines the mode leakage rate of the structure and has a tremendous influence on the structural *Q* factor. It is predictable that the hysteresis loop is going to vanish if *h* further increases unless a higher $I_{\mathrm{in}}$ is selected for the simulation.

Figure 6a shows the shift of nonlinear reflectance spectrums with the grating width-to-period ratio *δ* varying from 0.3 to 0.7. Likewise, other structural parameters are the same as those in Figure 2 and Figure 3, and $I_{\mathrm{in}}$ = 1.75 MW/cm^2^. The structure shows significant stability to the change of *δ* since the spectrum only shifts a little with *δ* varying from 0.4 to 0.65. Figure 6b shows the hysteresis loops with *δ* = 0.3, 0.5, and 0.7 at their corresponding working wavelengths, which are 1544.9, 1545.6, and 1545.9 nm, respectively. Even if *δ* extremely deviates away from the value of 0.5, the structure still possesses clear bistable characteristics. Similar to that in Figure 4b, the switching-off-to-on thresholds are well maintained at nearly 1.7 MW/cm^2^.

Finally, we would like to discuss the response time of the OB in the proposed graphene-based absorber, which is decided by both the graphene’s Kerr effect and the photon lifetime of the absorber. The photon lifetime $\tau_{R}$ is related to the *Q* factor of the absorber and can be calculated by $\tau_{R}=Q/2\pi v$ [57]. For the absorber in Figure 2 and Figure 3, the *Q* factor is 530 at a low incident intensity and close to 1000 at extremely high incident intensity when the absorption of graphene is saturated, so $\tau_{R}$ varies within the range from 434 fs to 821 fs. The photon lifetime $\tau_{R}$ is longer than the response time of graphene’s Kerr effect, which can be as short as 100 fs [57], as was previously measured. Therefore, the response time of OB in the proposed absorber is limited by $\tau_{R}$ of less than 821 fs.

The proposed perfect absorber can be easily fabricated with standard nanofabrication technology. The gold layer can be deposited onto a chromium-coated silicon substrate by magnetron sputtering. The silica layer can be deposited onto the gold layer by plasma-enhanced chemical vapor deposition (PECVD). The monolayer graphene can be obtained by mechanical exfoliation and can be transferred onto the top of the silica layer. A PMMA film can be spin coated onto the graphene, and the designed grating patterns can be formed by E-beam lithography [29,30]. We believe that the proposed bistable structure with low thresholds, high extinction ratio, ultrafast response time, and ultra-compact size could be of great applications in the field of all-optical-communication devices, such as all-optical switches, memory cells, transistors, etc.

## 4. Conclusions

This work demonstrates a graphene-based perfect absorber that possesses bistable characteristics in the near-IR band. By coupling monolayer graphene with dielectric grating patterns, the electric field intensity is highly enhanced within the graphene. The strong light-graphene interaction, together with the large nonlinear coefficient of graphene, highly contributes to low switching thresholds, which can be lower than 2.5 MW/cm^2^ according to the simulation. Meanwhile, the extinction ratio of bistable states exceeds 40 dB at certain incident intensities. The structure is compact and can be easily fabricated with standard nanofabrication technology, which is practical for designing and fabricating graphene-based bistable devices and all-optical switches in the near-IR band.

## Figures and Tables

**Figure 1 nanomaterials-13-00389-f001:**
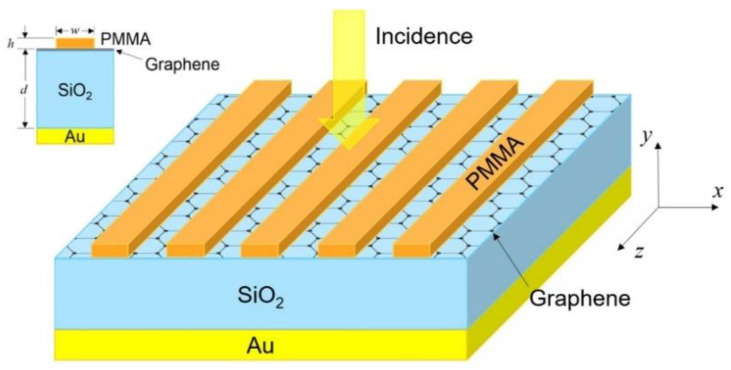
Schematic of the proposed graphene-based perfect absorber.

**Figure 2 nanomaterials-13-00389-f002:**
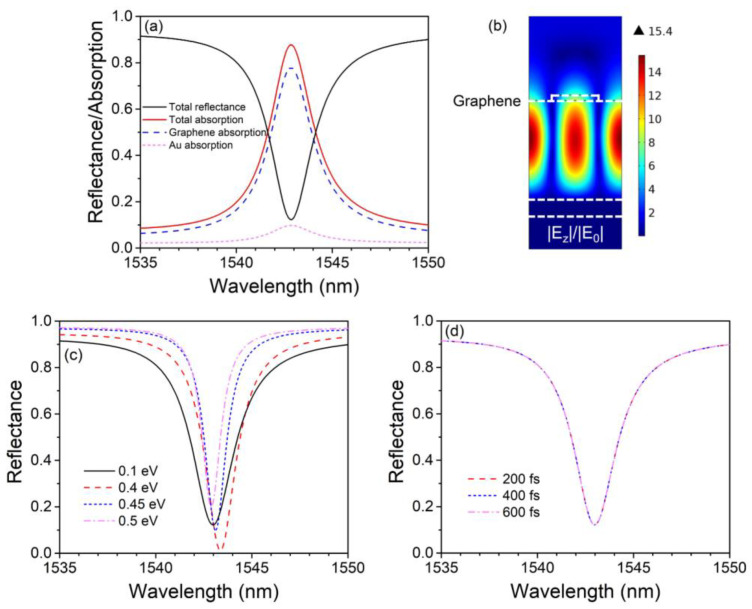
(**a**) Reflectance and absorption spectrums of the graphene-based perfect absorber. The black solid curve, red solid curve, blue dash curve, and magenta short dash curve plot the total reflectance, total absorption, graphene absorption, and gold absorption, respectively. (**b**) Distribution of normalized electric field at 1542.9 nm. (**c**) Reflectance spectrums with different Fermi levels. (**d**) Reflectance spectrums with different relaxation times.

**Figure 3 nanomaterials-13-00389-f003:**
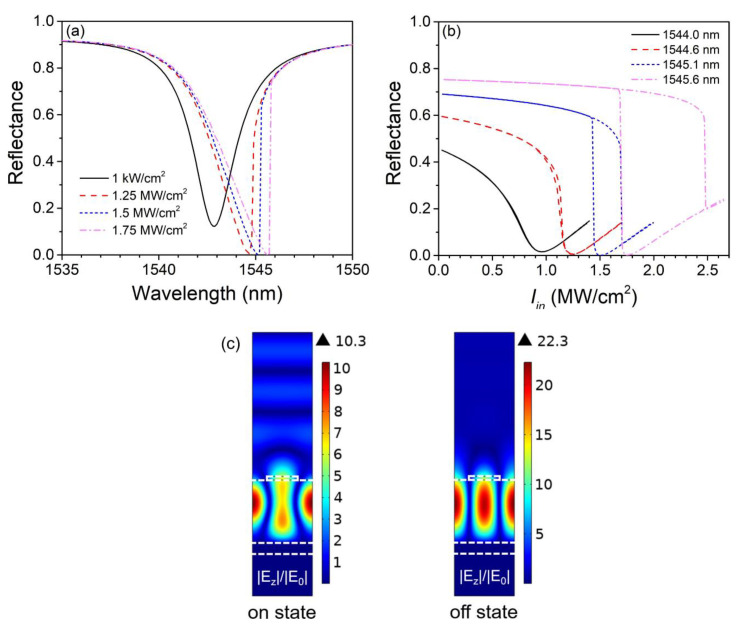
(**a**) Nonlinear reflectance spectrums at different incident intensities. (**b**) Hysteresis loops of reflectance at different working wavelengths. (**c**) Distribution of normalized electric field at the on and off states at 1545.6 nm, with $I_{\mathrm{in}}$ = 1.75 MW/cm^2^.

**Figure 4 nanomaterials-13-00389-f004:**
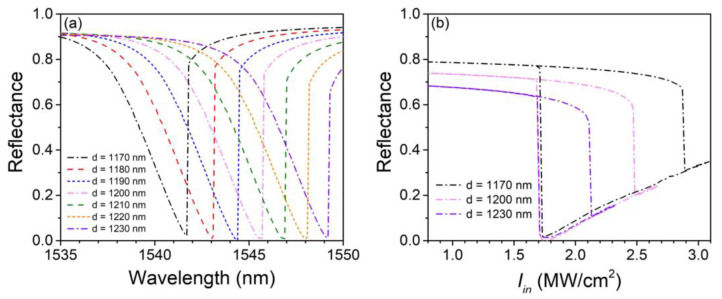
(**a**) Nonlinear reflectance spectrums with different silica layer thicknesses *d*. (**b**) Hysteresis loops with *d* = 1170, 1200, and 1230 nm, respectively.

**Figure 5 nanomaterials-13-00389-f005:**
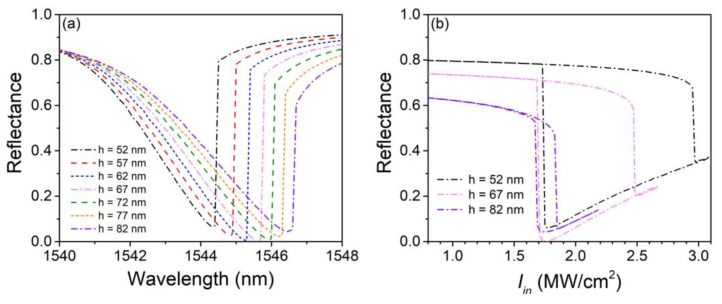
(**a**) Nonlinear reflectance spectrums with different grating thicknesses *h*. (**b**) Hysteresis loops with *h* = 52, 67, and 82 nm, respectively.

**Figure 6 nanomaterials-13-00389-f006:**
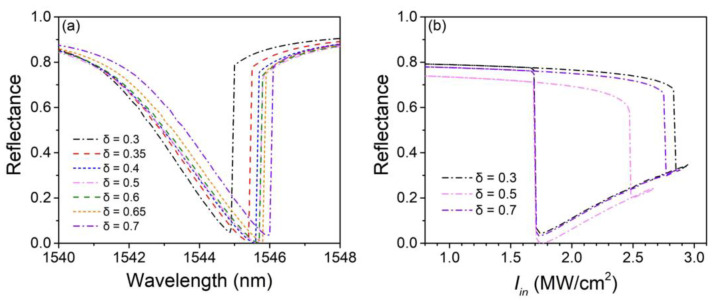
(**a**) Nonlinear reflectance spectrums with different grating width-to-period ratio *δ*. (**b**) Hysteresis loops with *δ* = 0.3, 0.5, and 0.7, respectively.

**Table 1 nanomaterials-13-00389-t001:** Comparison of this work with previous works.

Reference	Waveband	Threshold	Extinction Ratio
[43]	NIR	0.1 GW/cm^2^	>30 dB
[44]	NIR	0.13 GW/cm^2^	/
[45]	NIR	0.1 TW/cm^2^	/
[46]	THz	0.09 MW/cm^2^	≈13 dB
[47]	THz	0.01 MW/cm^2^	<10 dB
This work	NIR	1.43 MW/cm^2^	>40 dB

## Data Availability

Not applicable.

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
