# Peer review of "Low-Threshold and High-Extinction-Ratio Optical Bistability within a Graphene-Based Perfect Absorber"

_nanomaterials, 2023, doi:10.3390/nano13030389_

Round 1

Reviewer 1 Report

In this work the nonlinear effect of optical bistability in a layered structure with graphene sheet (graphene monolayer) is investigated. Monolayer graphene is sandwiched between a 1D polymethy1-methacrylate (PMMA) grating and a silica layer deposited onto a gold substrate. Owing to the strong light-graphene interaction (due to the resonance light localization in the silica layer) as well as the large nonlinear coefficient of graphene, the structure possesses bistable characteristics. The dependence of nonlinear reflectance on the structure parameters (grating thickness and width-to-period ratio, thickness of the silica layer) is investigated. The authors also discussed the response time of the optical bistability in the proposed graphene-based absorber, which is decided by both the graphene’s Kerr effect and the photon lifetime of the absorber. Obtained results will have an impact on further research dedicated to the development of efficient plasmonic structures.   

I recommend this paper be published in  NANOMATERIALS after revision.

My comments and suggestions:

- The authors should add a discussion about the importance of using a grating. What is it for? Why not replace it with a PMMA layer (film) of constant thickness?

- The grating period is comparable to the wavelength of electromagnetic radiation. In this case, the manifestation of diffraction effects should be expected. Why is this not taken into account in the study?

- It is necessary to compare the obtained values of the response time (less than 821 fs) and switching-on-to-off (switching-off-to-on) thresholds (1.7 MW/cm^2) with the known values presented in the works of other researchers.

- line 47: It is necessary to clarify what the abbreviation ‘OB’ means.

- line 93: It is necessary to specify the values of alpha_s and alpha_ns.

- line 219: Why do authors consider the value delta=0.5 as ‘the optimized value’?

- line 239: ‘The photon lifetime τ_R is longer than the response time of graphene’s Kerr effect which can be as short as 100 fs, as was previously measured.’ There is no link to the article that provides the value of the response time of graphene's Kerr effect.

- line 246: ‘A PMMA film can be spin coated onto the graphene and the designed grating patterns can be formed by E-beam lithography.’  Clarification is needed to this sentence: how can grating patterns be formed by E-beam lithography without damaging the graphene?

Reviewer 2 Report

I suggest major revision for this paper and it has novelty. Here are my comments:

1-English grammar needs to checked and revised in the whole doc.

2-A comparison table should be added before the conclusion section. The authors should compare this work with at least 5 previously published ones in the absorber area in 2021-2022.

3-Reflection spectra for different chemical potential values should be given in one figure. Please also present good physical description behind the spectra.

4-Reflection spectra for different relaxation time values should be given in one figure. Please also present good physical description behind the spectra.

5-The paper can be cited in the revision: Graphene-based dual-functional chiral metamirror composed of complementary 90° rotated U-shaped resonator arrays and its equivalent circuit model | Scientific Reports (nature.com)

Reviewer 3 Report

The paper presents the simulation results for  a kind of graphene-based perfect absorber possessing optical bistability in the near-IR band. The potential characteristics of the device are demonstrated. There are however two main aspects that have not been taken into account. The both are related to the power densities of 2.5 MW/cm2 mentioned in the paper. It is a large optical power density involving the absorption of 10^24 photos per second in the graphene-based structure.  This makes huge problems with heating the device and without the adequate cooling the graphene layer will be instantaneously damaged. The second aspect is dealing with the optical source capable to emit such huge numbers of photons.

Author Response

We thank the reviewer for this comment. The power density needed in the optical bistability is really a critical issue as for real applications. Fortunately, graphene has excellent thermal conductivity and high melting point. Previous experiments have shown that graphene can withstand very high optical power density. For example, graphene can withstand peak power intensity over 1.7 TW/cm2 (Yoshikawa, Naotaka, Tomohiro Tamaya, and Koichiro Tanaka. "High-harmonic generation in graphene enhanced by elliptically polarized light excitation." Science 356.6339 (2017): 736-738). Meanwhile, the threshold of optical bistability in our structure can be dramatically decreased if we increase the Q factor of the absorber (currently, the Q factor is lower than 1000). In addition, we compared the optical bistability threshold in our structure with that in other graphene-based structures (table 1, in the revised manuscript), our threshold is much lower than that of other reported results in the near IR range.

As for the optical source, we can use a fiber amplifier to amplify the laser output from a tunable laser, and the amplified optical power can reach 1W. If we focus the laser beam in a spot with size small than 100 μm2, then the light intensity will over 1 MW/cm2.

Round 2

Reviewer 2 Report

The paper is acceptable now.